# Ebola virus sequesters IRF3 in viral inclusion bodies to evade host antiviral immunity

Lin Zhu[1]*, Jing Jin[2], Tingting Wang[2], Yong Hu[1], Hainan Liu[1], Ting Gao[1], Qincai Dong[1], Yanwen Jin[1], Ping Li[1], Zijing Liu[1], Yi Huang[3]*, Xuan Liu[1]*, Cheng Cao[1]*

[1]Institute of Biotechnology, Academy of Military Medical Sciences, Beijing, China; [2]Institute of Physical Science and Information Technology, Anhui University, Hefei, China; [3]Wuhan Institute of Virology, Chinese Academy of Sciences, Wuhan, China

**\*For correspondence:**
zhul627@gmail.com (LZ);
hy@wh.iov.cn (YH);
liux931932@163.com (XL);
caoc@nic.bmi.ac.cn (CC)

**Competing interest:** The authors declare that no competing interests exist.

**Abstract** Viral inclusion bodies (IBs) commonly form during the replication of Ebola virus (EBOV) in infected cells, but their role in viral immune evasion has rarely been explored. Here, we found that interferon regulatory factor 3 (IRF3), but not TANK-binding kinase 1 (TBK1) or IκB kinase epsilon (IKKε), was recruited and sequestered in viral IBs when the cells were infected by EBOV transcription- and replication-competent virus-like particles (trVLPs). Nucleoprotein/virion protein 35 (VP35)-induced IBs formation was critical for IRF3 recruitment and sequestration, probably through interaction with STING. Consequently, the association of TBK1 and IRF3, which plays a vital role in type I interferon (IFN-I) induction, was blocked by EBOV trVLPs infection. Additionally, IRF3 phosphorylation and nuclear translocation induced by Sendai virus or poly(I:C) stimulation were suppressed by EBOV trVLPs. Furthermore, downregulation of STING significantly attenuated VP35-induced IRF3 accumulation in IBs. Coexpression of the viral proteins by which IB-like structures formed was much more potent in antagonizing IFN-I than expression of the IFN-I antagonist VP35 alone. These results suggested a novel immune evasion mechanism by which EBOV evades host innate immunity.

## eLife assessment

This study explores how Ebola virus evades human immune responses. The study reports a potential new mechanism wherein Ebola virus traps human IRF3, a key transcription factor involved in immune signaling, into virus-produced "inclusion bodies". The topic is **important**, the paper has many merits, and the biochemical assays are **solid**. However, the current data do not clearly explain the relationship between the VP35 protein and IRF3.

## Introduction

Ebola virus (EBOV) disease is the deadliest infectious disease caused by infection with EBOV, an enveloped, nonsegmented negative-sense RNA virus (*Feldmann et al., 2003*). The 19 kb viral genome comprises seven genes encoding the nucleoprotein (NP), virion protein 35 (VP35), VP40, glycoprotein (GP), VP30, VP24, and RNA-dependent RNA polymerase (L) (*Mahanty and Bray, 2004*). Inclusion bodies (IBs) that form in EBOV-infected cells are specialized intracellular compartments that serve as sites for EBOV replication and the generation of progeny viral RNPs (*Hoenen et al., 2012*; *Nanbo et al., 2013*). In IBs, the EBOV genome is replicated and transcribed by viral polymerase complexes (*Misasi and Sullivan, 2014*). VP35 serves as a cofactor of RNA-dependent RNA polymerase and

contributes to viral replication by homo-oligomerization through a coiled-coil domain (*Reid et al., 2005*) as well as through its phosphorylation and ubiquitination, which was recently discovered (*van Tol et al., 2022*; *Zhu et al., 2020*).

Innate interferon responses constitute the first lines of host defense against viral infection. Retinoic acid-inducible gene I (RIG-I)-like receptors (RLRs), including RIG-I and melanoma differentiation-associated protein 5, play pivotal roles in the response to RNA virus infection. After the recognition of RNA virus infection, RIG-I is recruited to the mitochondrial antiviral adaptor protein (MAVS) through the caspase activation and recruitment domain. The activation of MAVS recruits multiple downstream signaling components to mitochondria, leading to the activation of inhibitor of κ-B kinase ε (IKKε) and TANK-binding kinase 1 (TBK1), which in turn phosphorylate IFN regulatory factor 3 (IRF3). Phosphorylated IRF3 forms a dimer that translocates to the nucleus, where it activates the transcription of type I interferon (IFN-I) genes (*Fitzgerald et al., 2003*; *Liu et al., 2015*).

To promote viral replication and persistence, viruses have evolved various strategies to evade or subvert host antiviral responses. For example, severe fever with thrombocytopenia syndrome virus (SFTSV) has developed a mechanism to evade host immune responses through the interaction between nonstructural proteins and IFN-I induction proteins, including TBK1, IRF3, and IRF7 (*Hong et al., 2019*; *Lee and Shin, 2021*; *Ning et al., 2014*; *Wu et al., 2014*), sequestering them inside SFTSV-induced cytoplasmic structures known as IBs. In addition to inhibiting IFN-I induction, SFTSV nonstructural proteins can hijack STAT1 and STAT2 in IBs to suppress IFN-I signaling (*Ning et al., 2015*). These studies highlight the role of viral IBs as virus-built 'jails' that sequester some crucial host factors and interfere with the corresponding cellular processes.

EBOV uses various approaches to evade the host immune response, including antagonizing IFN production, inhibiting IFN signaling, and enhancing IFN resistance (*Basler et al., 2000*; *McCarthy et al., 2016*; *Reid et al., 2006*). VP35 is an IFN-I inhibitor that antagonizes host innate immunity by interacting with TBK1 and IKKε (*Basler et al., 2003*; *Prins et al., 2009*), suppressing RNA silencing and inhibiting dendritic cell maturation (*Haasnoot et al., 2007*; *Yen et al., 2014*). Here, we report that viral IBs in EBOV transcription- and replication-competent virus-like particle (trVLP)-infected cells appear to play a role in immune evasion by sequestering IRF3 into IBs and preventing the interaction of IRF3 with TBK1 and IKKε.

## Results

### IRF3 is hijacked into cytoplasmic IBs in EBOV transcription and replication-competent virus-like particles infected cells

When HepG2 cells were infected with EBOV trVLPs (*Hoenen et al., 2014*), which authentically model the complete virus life cycle, IBs with a unique structure and viral particles formed in the cytoplasm (*Figure 1—figure supplement 1A, B*). Surprisingly, we found that a substantial percentage of endogenous IRF3 was trapped in viral IBs in EBOV trVLP-infected cells with large IBs (*Figure 1A, B*), while no detectable TBK1 or IKKε, the essential upstream components of IRF3 signaling (*Fitzgerald et al., 2003*), was sequestered in the viral IBs (*Figure 1C–F*). These results suggested that IRF3 was specifically compartmentalized in viral IBs, and this compartmentalization spatially isolated IRF3 from its upstream activators TBK1 and IKKε.

The sequestration of IRF3 in IBs was further investigated at different hours post infection (hpi) of EBOV trVLPs. Detectable IRF3 puncta colocalized with viral proteins were apparent at 36 hpi in infected cells and correlated significantly with the size and shape of the viral IBs (*Figure 2A, B*). As the size of IBs increased at 48 hpi, nearly all IRF3 colocalized with viral IBs, whereas the IRF3 distribution was completely different in the uninfected cells nearby (*Figure 2A, B*). Using a fluorophore line of interest analysis, we assessed the intensity profiles of cytoplasmic IRF3 intensity in IBs as well as the increase in the diameter of the aggregates (*Figure 2C*). As infection proceeded, the intensity of the IRF3 signal in the puncta increased as the level of cytoplasmic-dispersed IRF3 decreased (*Figure 2A*), indicating that the total amount of IRF3 in the cells did not dramatically change during infection (*Figure 2D, E*) and that only its subcellular localization changed. Taken together, the results above showed that IRF3, but not TBK1 or IKKε, was sequestered in viral IBs.

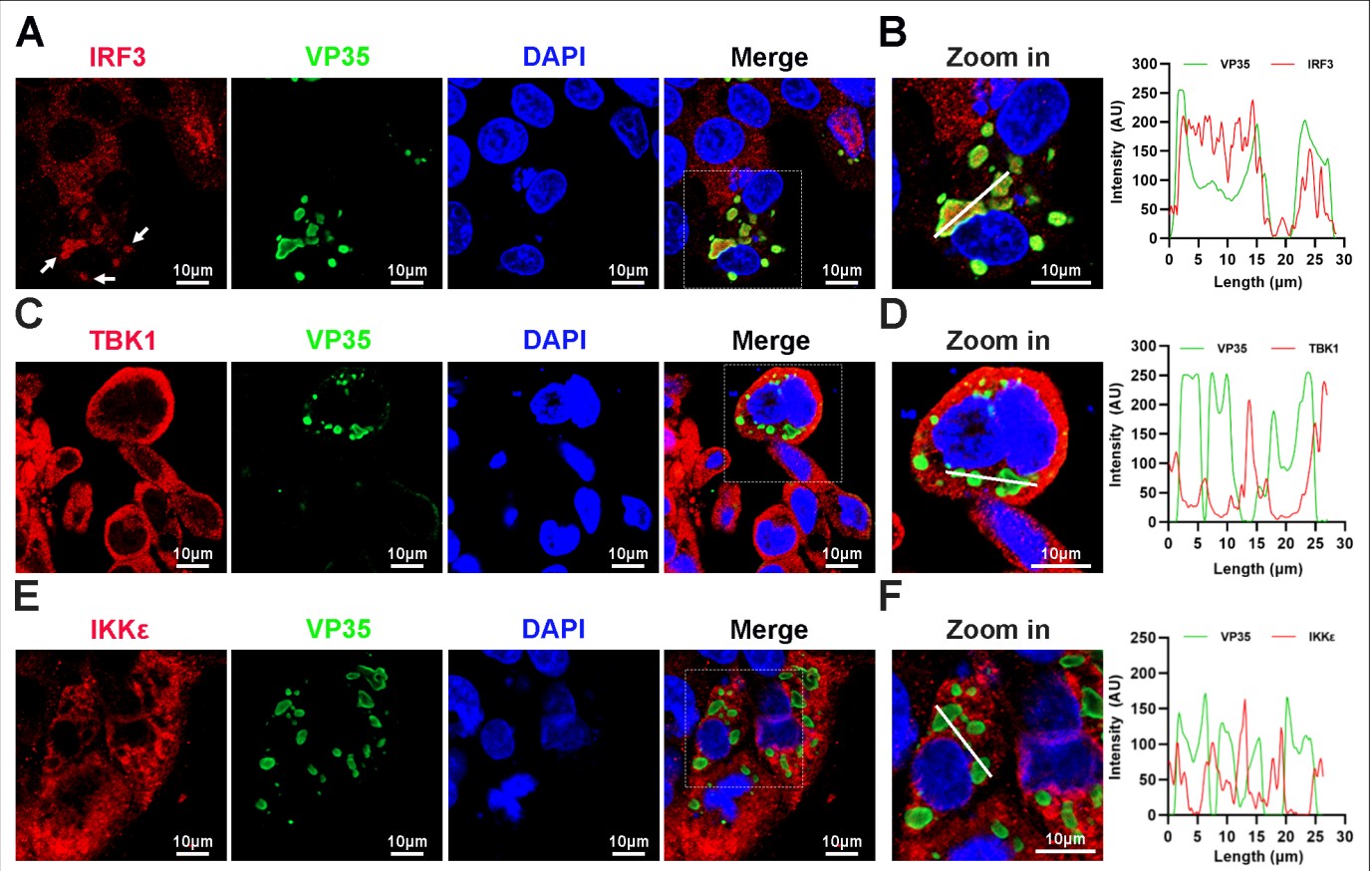

**Figure 1.** Interferon regulatory factor 3 (IRF3), but not TANK-binding kinase 1 (TBK1) and I κ B kinase epsilon (IKKε), is sequestered into viral inclusion bodies (IBs) upon Ebola virus (EBOV) transcription- and replication-competent virus-like particles (trVLPs) infection. (**A**) HepG2 cells infected with the EBOV trVLPs were immunostained with anti-IRF3 (red) and anti-VP35 (green) antibodies. Nuclei were stained with DAPI (4′,6-diamidino-2phenylindole; blue), and images were obtained using a Zeiss LSM 800 Meta confocal microscope. White arrows: IRF3 in IBs. (**B**) The left panel shows a magnified image of the IBs boxed in the merged panel of (**A**). The graphs (right panel) show the fluorescent intensity profiles along the indicated white lines drawn across one or more IBs. (**C, E**) HepG2 cells infected with the EBOV trVLPs were immunostained with anti-TBK1 (red in (**C**)) or anti-IKKε (red in (**E**)) and anti-VP35 (green in (**C, E**)) antibodies. Nuclei were stained with DAPI (blue), and images were obtained using a Zeiss LSM 800 Meta confocal microscope. Scale bar, 10 μm. (**D, F**) The left panel shows a magnified image of the IBs boxed in the merged panel shown in (**C**) and (**E**). The graphs (right panel) show the fluorescent intensity profiles along the indicated white lines drawn across one or more IBs.

The online version of this article includes the following source data and figure supplement(s) for figure 1:

**Source data 1.** Numerical data for *Figure 1B*.

**Source data 2.** Numerical data for *Figure 1D*.

**Source data 3.** Numerical data for *Figure 1F*.

**Figure supplement 1.** Transmission electron microscopy and immunofluorescence detection of Ebola virus (EBOV) transcription- and replication-competent virus-like particles (trVLPs) and inclusion bodies (IBs).

## EBOV trVLPs infection attenuates the TBK1–IRF3 association and IRF3 nuclear translocation

Upon virus infection, IRF3, as a critical transcription factor in the IFN induction pathway, can be phosphorylated and activated by TBK1, and then phosphorylated IRF3 translocates from the cytoplasm into the nucleus, eliciting the expression of antiviral IFNs. Given the sequestration of IRF3 by EBOV trVLPs in IBs, the TBK1–IRF3 association in EBOV trVLP-infected cells was assessed by an in situ Duolink proximity ligation assay (PLA). Cytoplasmic complexes consisting of endogenous TBK1 with IRF3 (the red signal) were observed in HepG2 cells treated with poly(I:C), which induces the activation of the RIG-I signal cascade and IRF3 phosphorylation, and poly(I:C)-induced TBK1:IRF3 complexes were significantly reduced by EBOV trVLPs infection (*Figure 3A, B*). Decreased TBK1–IRF3 association

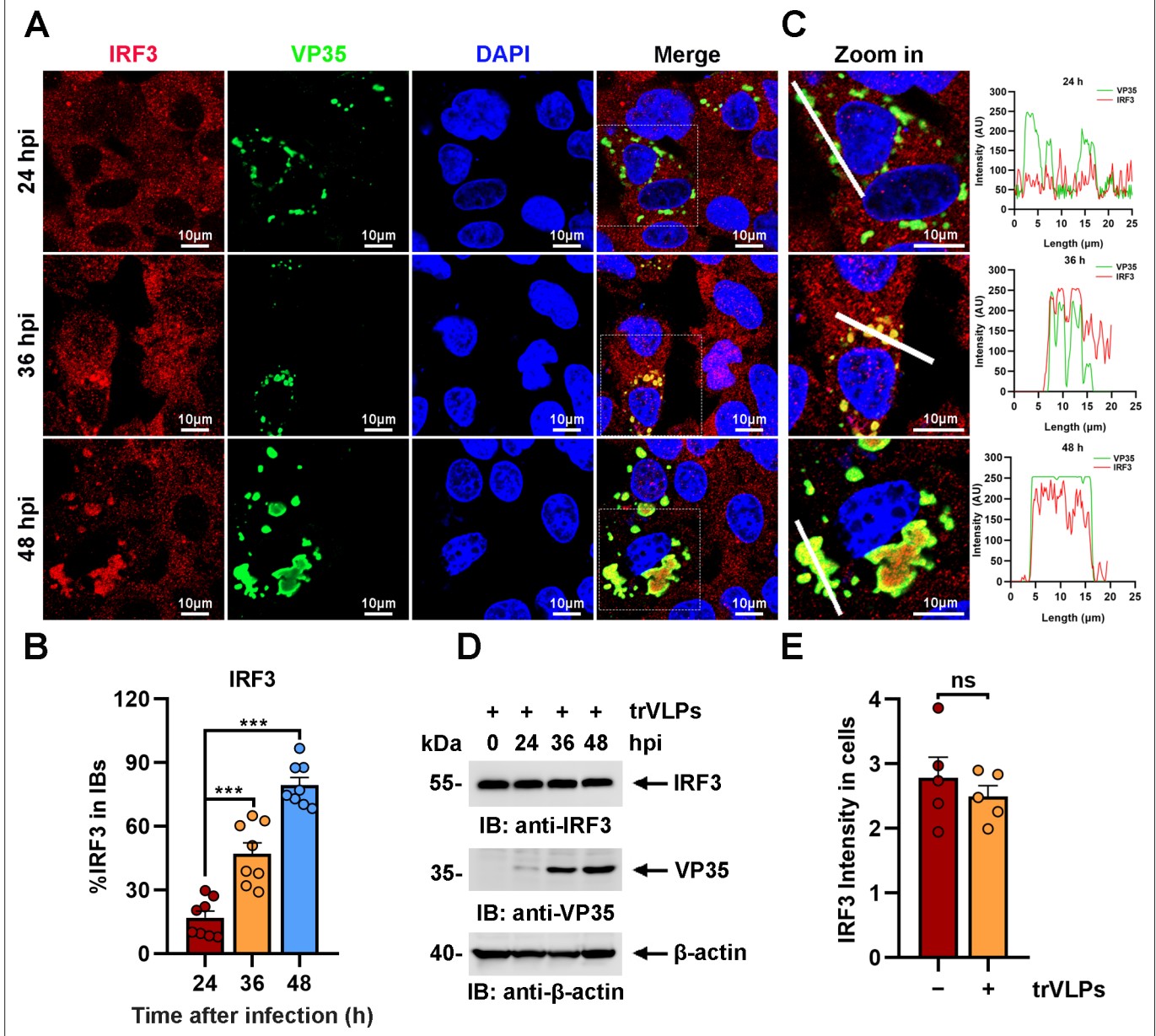

**Figure 2.** Ebola virus (EBOV) transcription- and replication-competent virus-like particles (trVLPs) induce the recruitment of interferon regulatory factor 3 (IRF3) into intracytoplasmic inclusion bodies (IBs). (**A**) HepG2 cells were infected with EBOV trVLPs. At the indicated time points after infection, cells were fixed and immunostained with anti-IRF3 (red) and anti-VP35 (green) antibodies. Nuclei were stained with DAPI (blue), and images were obtained using a Zeiss LSM 800 Meta confocal microscope. Scale bar, 10 µm. The data from two independent replicates are presented. (**B**) The percentage of IRF3 distribution in IBs at different time points in cells infected with EBOV trVLPs (**A**) was analyzed using the R programming language. The intensity of IRF3 in eight cells from two independent assays is presented as the mean ± standard error of the mean (SEM; $n = 8$; ***$p < 0.001$). (**C**) The left panel shows a magnified image of the IBs boxed in the merged panel shown in (**A**). The graphs (right panel) show the fluorescent intensity profiles along the indicated white lines drawn across one or more IBs. (**D**) IRF3 levels in HepG2 cells infected with EBOV trVLPs were analyzed by immunoblotting with an anti-IRF3 antibody at the indicated hours post infection (hpi). (**E**) The IRF3 intensity in cells infected with or without EBOV trVLPs for 48 hr (the lower panel of (**A**)) was analyzed using ImageJ software. Differences between the two groups were evaluated using a two-sided unpaired Student's *t*-test. The intensity of IRF3 in five cells from two independent assays is presented as the mean ± SEM ($n = 5$; ns, not significant).

The online version of this article includes the following source data for figure 2:

**Source data 1.** Numerical data for *Figure 2B*.

**Source data 2.** Numerical data for *Figure 2C*.

**Source data 3.** Raw image for *Figure 2D*.

**Source data 4.** Numerical data for *Figure 2E*.

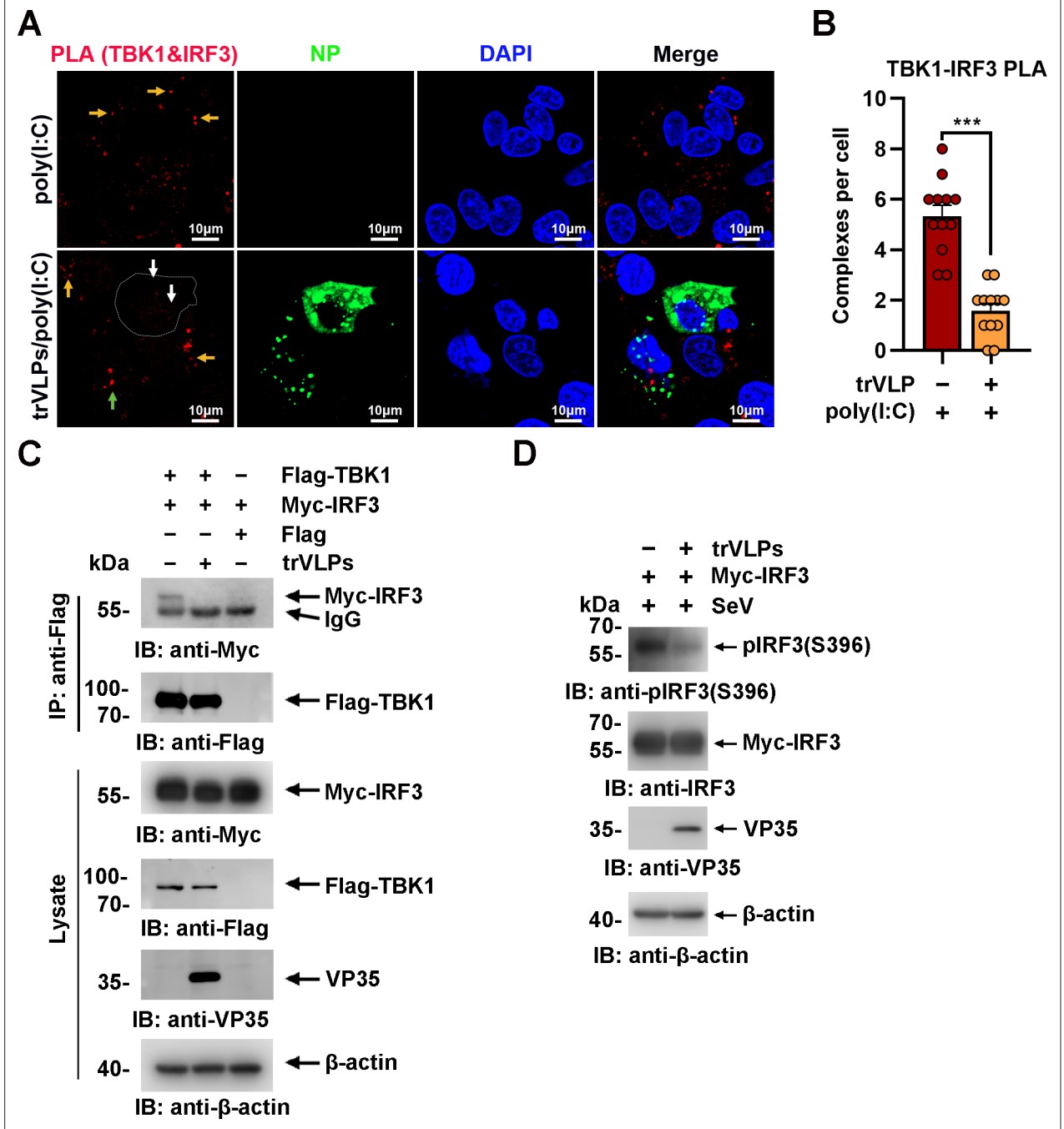

**Figure 3.** Ebola virus (EBOV) transcription- and replication-competent virus-like particles (trVLPs) inhibit interferon regulatory factor 3 (IRF3) activation. (**A**) HepG2 cells were infected with or without the EBOV trVLPs. Thirty-six hours after infection, the cells were treated with or without 5 μg/ml poly(I:C) for 12 hr and then subjected to in situ proximity ligation assay (PLA) with anti-TANK-binding kinase 1 (TBK1) and anti-IRF3 antibodies and immunostaining with an anti-NP antibody (green). Nuclei were stained with DAPI (blue), and images were obtained using a Zeiss LSM 800 Meta confocal microscope. Arrows: white arrows indicate TBK1–IRF3 complexes in trVLP-infected cells, and yellow and green arrows indicate TBK1–IRF3 complexes in uninfected and infected cells with small inclusion bodies (IBs), respectively. Scale bar, 10 μm. (**B**) The signal for the PLA complex in each cell in (**A**) was counted from at least 12 cells and is presented as the mean ± standard error of the mean (SEM, ***p<0.001). (**C**) Lysates of HEK293 cells cotransfected with or without the EBOV minigenome (p0) and the indicated plasmids were subjected to anti-Flag immunoprecipitation and analyzed by immunoblotting. (**D**) HEK293 cells were cotransfected with or without the EBOV minigenome (p0) and Myc-IRF3 plasmids. Thirty-six hours after transfection, the cells were infected with Sendai virus (SeV) at an MOI (multiplicity of infection) of 2 for 12 hr, and the phosphorylation of IRF3 was analyzed by immunoblotting with an anti-IRF3-S396 antibody.

The online version of this article includes the following source data for figure 3:

*Figure 3 continued on next page*

*Figure 3 continued*

**Source data 1.** Numerical data for *Figure 3B*.

**Source data 2.** Raw image for *Figure 3C*.

**Source data 3.** Raw image for *Figure 3D*.

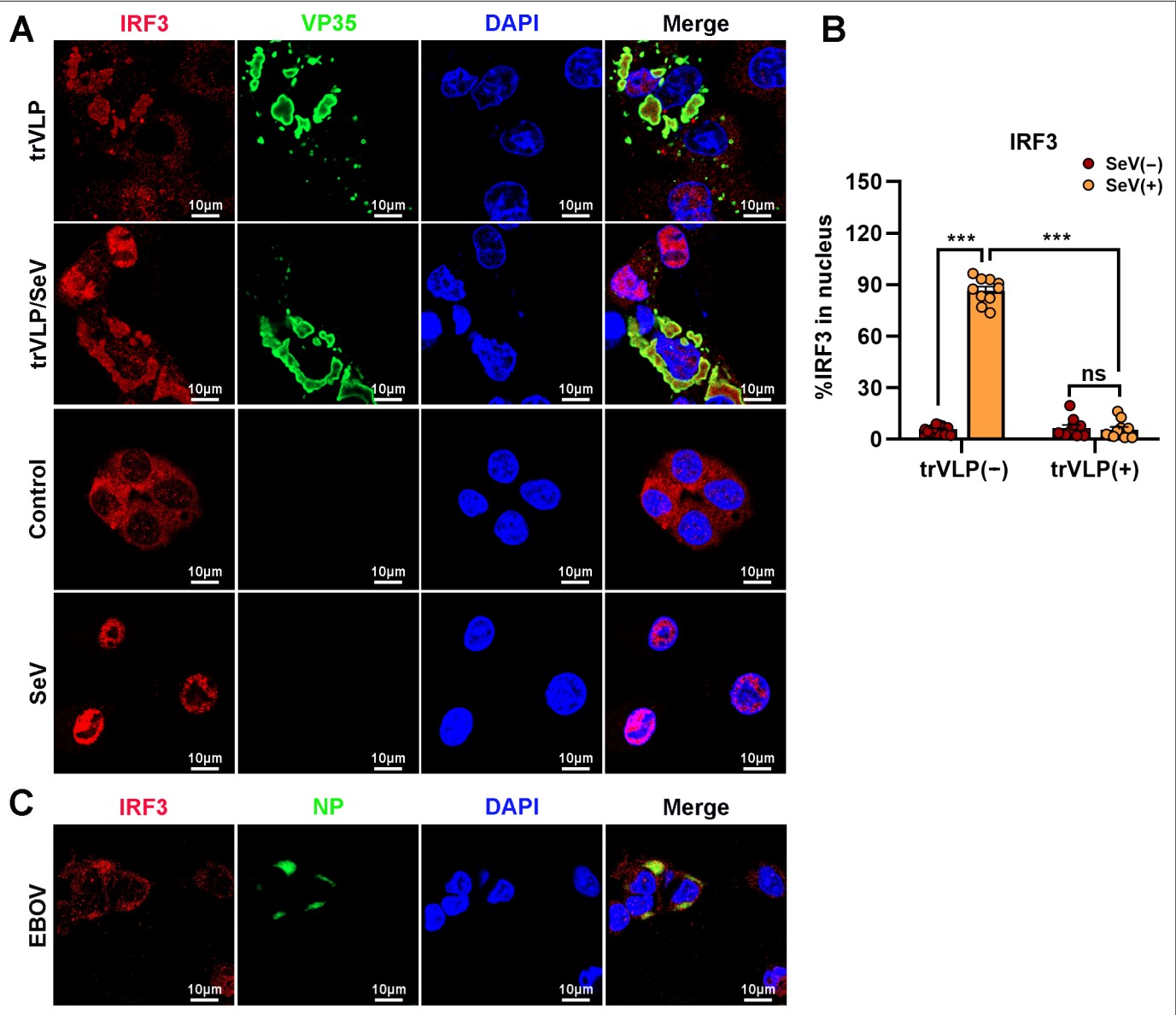

**Figure 4.** Ebola virus (EBOV) transcription- and replication-competent virus-like particles (trVLPs) inhibit nuclear translocation of interferon regulatory factor 3 (IRF3). (**A**) HepG2 cells were infected with or without the EBOV trVLPs for 36 hr, and the cells were infected with or without Sendai virus (SeV) at an MOI of 2 for another 12 hr. The cells were then fixed and immunostained with anti-IRF3 (red) and anti-VP35 (green) antibodies. Nuclei were stained with DAPI (blue), and images were obtained using a Zeiss LSM 800 Meta confocal microscope. Scale bar, 10 μm. (**B**) The percentage of IRF3 nuclear distribution in (**A**) was analyzed using ImageJ software. The ratio of IRF3 distribution in ten cells from two independent assays is presented as the mean ± standard error of the mean (SEM; ns, not significant, ***p < 0.001). (**C**) HepG2 cells infected with live EBOV (MOI = 10) for 72 hr were immunostained with anti-IRF3 (red) and anti-NP (green) antibodies. Nuclei were stained with DAPI (blue), and images were obtained using a Zeiss LSM 800 Meta confocal microscope. Scale bar, 10 μm.

The online version of this article includes the following source data for figure 4:

**Source data 1.** Numerical data for *Figure 4B*.

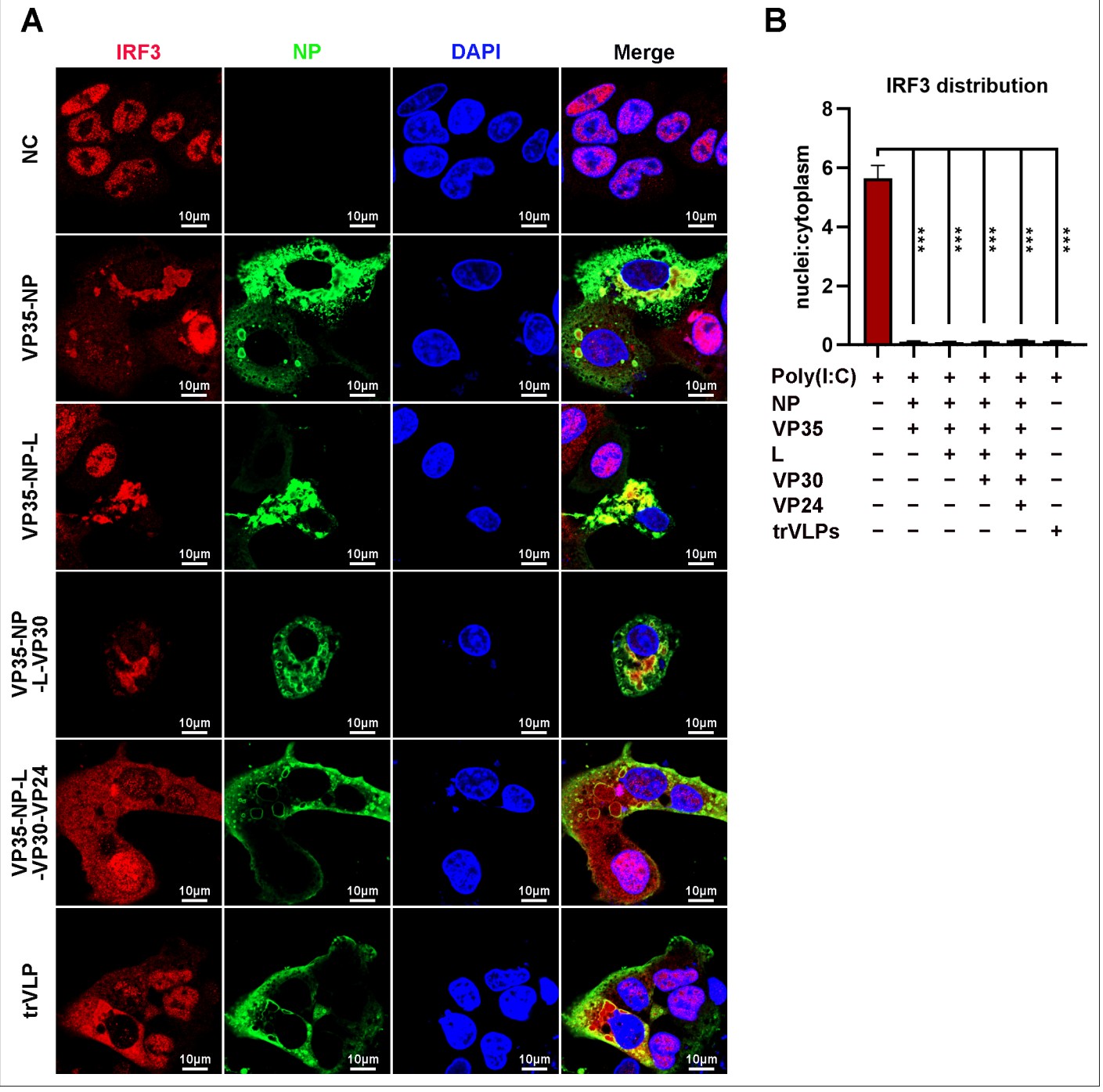

**Figure 5.** Ebola virus (EBOV) nucleoprotein (NP) and virion protein 35 (VP35) play an important role in sequestering interferon regulatory factor 3 (IRF3) into inclusion bodies (IBs). (**A**) HepG2 cells were transfected with the indicated plasmids for 36 hr, and the cells were treated with 5 µg/ml poly(I:C) for another 12 hr. Then, the cells were fixed and immunostained with anti-IRF3 (red) and anti-NP (green) antibodies. Nuclei were stained with DAPI (blue), and images were obtained using a Zeiss LSM 800 Meta confocal microscope. Scale bar, 10 µm. (**B**) The nuclear/cytoplasmic distribution of IRF3 in (**A**) was analyzed by ImageJ software. Differences between the two groups were evaluated using a two-sided unpaired Student's *t*-test. The ratio of IRF3 distribution in at least five cells from two independent assays is presented as the mean ± standard error of the mean (SEM; *n* = 5; ***p < 0.001).

The online version of this article includes the following source data and figure supplement(s) for figure 5:

**Source data 1.** Numerical data for *Figure 5B*.

**Figure supplement 1.** Neither virion protein 35 (VP35) nor nucleoprotein (NP) interacts directly with interferon regulatory factor 3 (IRF3) in cells.

**Figure supplement 1—source data 1.** Raw image for *Figure 5—figure supplement 1A*.

*Figure 5 continued on next page*

was further demonstrated by immunoprecipitation (*Figure 3C*). Moreover, as shown in *Figure 3D*, Sendai virus (SeV) infection-induced IRF3 phosphorylation and nuclear translocation were significantly inhibited by EBOV trVLPs (*Figure 3D* and *Figure 4A, B*). Importantly, IRF3 was also recruited into IB-like compartments in the cytoplasm in the cells infected with live EBOV (*Figure 4C*). These data collectively suggested that the EBOV-mediated sequestration of IRF3 in IBs blocks IRF3 phosphorylation and nuclear translocation in the TBK1–IRF3 signaling cascade, which is critical for IFN induction.

## IB-like structures formed by the viral proteins VP35 and NP play a key role in inducing IRF3 sequestration

Ectopic expression of NP alone (*Noda et al., 2007*) or NP and the VP35 protein (*Noda et al., 2011*) in cells was sufficient to form IB-like structures. To investigate the viral protein(s) involved in the sequestration of IRF3 in IBs, HepG2 cells were transiently transfected with plasmids encoding NP/VP35, NP/VP35/L, NP/VP35/L/VP30, NP/VP35/L/VP30/VP24, or NP/VP35/L/VP30/vRNA-RLuc/T7 and stained with anti-IRF3 and anti-NP at 48 hpi. Coexpression of NP and VP35 resulted in substantial sequestration of IRF3 in the IB-like structure, which in turn resulted in a significant reduction of IRF3 in the nucleus, as observed in the cells transfected with vectors only and treated with poly(I:C) (*Figure 5A, B*). Little if any VP35 or NP was demonstrated to interact with IRF3 by immunoprecipitation (*Figure 5—figure supplement 1A, B*). Compared to NP/VP35 coexpression, the presence of protein L, VP30 and VP24 showed little, if any, effects on IB-like structure formation, IRF3 sequestration and nuclear IRF3 levels (*Figure 5A, B* and *Figure 5—figure supplement 2A, B*). These results suggested that IB-like structures as well as VP35 expression were indispensable for IRF3 sequestration.

## VP35:STING interactions play an important role in isolating IRF3 into viral IBs

TBK1 and IKKε were spatially separated from VP35 upon infection by EBOV trVLPs (*Figure 1C, E*), and IRF3 itself was demonstrated not to interact with VP35 and NP (*Figure 5—figure supplement 1A, B*), implying that other IRF3-interacting proteins might be involved in IRF3 sequestration in IBs upon viral infection. Stimulator of IFN genes (STING), an endoplasmic reticulum adaptor associated with IRF3 (*Petrasek et al., 2013*), was observed to interact with VP35 (*Figure 6A*) and be recruited into IBs when the cells were infected by EBOV trVLPs (*Figure 6B, C*). A substantial portion of STING was found to be recruited into IBs at 36 hpi in EBOV trVLP-infected cells (*Figure 6D, E* and *Figure 6—figure supplement 1*). STING knockdown by small interfering RNA (siRNA) inhibited IRF3 sequestration in viral IBs (*Figure 6F, G*). These results suggested that STING played important roles in the sequestration of IRF3 in viral IBs, possibly by interacting with VP35.

## Viral IB-induced IRF3 sequestration suppresses IFN-β production

EBOV trVLPs could hijack IRF3 and sequester IRF3 into IBs and thus block the nuclear translocation of IRF3, which suggested that EBOV trVLPs may suppress IRF3-driven IFN-β production. As reported previously (*Basler et al., 2000*), expression of VP35 (*Figure 7A*), but not NP, resulted in a mild inhibition of SeV-induced IFN-β-Luc expression (*Figure 7B*). Coexpression of VP35 and NP, which led to the formation of IBs and the sequestration of IRF3 (*Figure 5A*), suppressed IFN-β-Luc expression much more potently than VP35 expression alone (*Figure 7B*). Coexpression of NP/VP35/L/VP30 was more potent in the inhibition of SeV-induced IFN-β-Luc expression than NP/VP35 (*Figure 7B*). Moreover, coexpression of NP/VP35/VP30/L almost completely suppressed poly(I:C)-induced IFN-β transcription (*Figure 7C*). *IRF3* depletion showed little, if any, effect on IFN-β transcription upon NP/VP35/L/VP30 coexpression (*Figure 7C* and *Figure 7—figure supplement 1A*), which suggested that NP/VP35/L/VP30 coexpression was similarly powerful as *IRF3* depletion in antagonizing IFN-β expression. In wild-type cells but not *IRF3*-depleted cells, the coexpression of NP/VP35/L/VP30 had a significantly greater ability to inhibit SeV-induced transcription of IFN-β downstream genes, such as CXCL10, ISG15, and ISG56, than VP35 expression alone (*Figure 7D–F*). These results strongly suggested that the sequestration of IRF3 in viral IBs was substantially more powerful than that upon VP35 expression.

We next assessed the effect of IRF3 hijacking and sequestration by viral IBs on EBOV trVLPs replication. Compared with wild-type cells, *IRF3* depletion showed little, if any, effect on EBOV replication, as indicated by luciferase activity, suggesting that trVLPs efficiently blocked IRF3 signaling (*Figure 7G* and *Figure 7—figure supplement 1B*). Moreover, the overexpression of IRF3/5D (a phospho-mimic

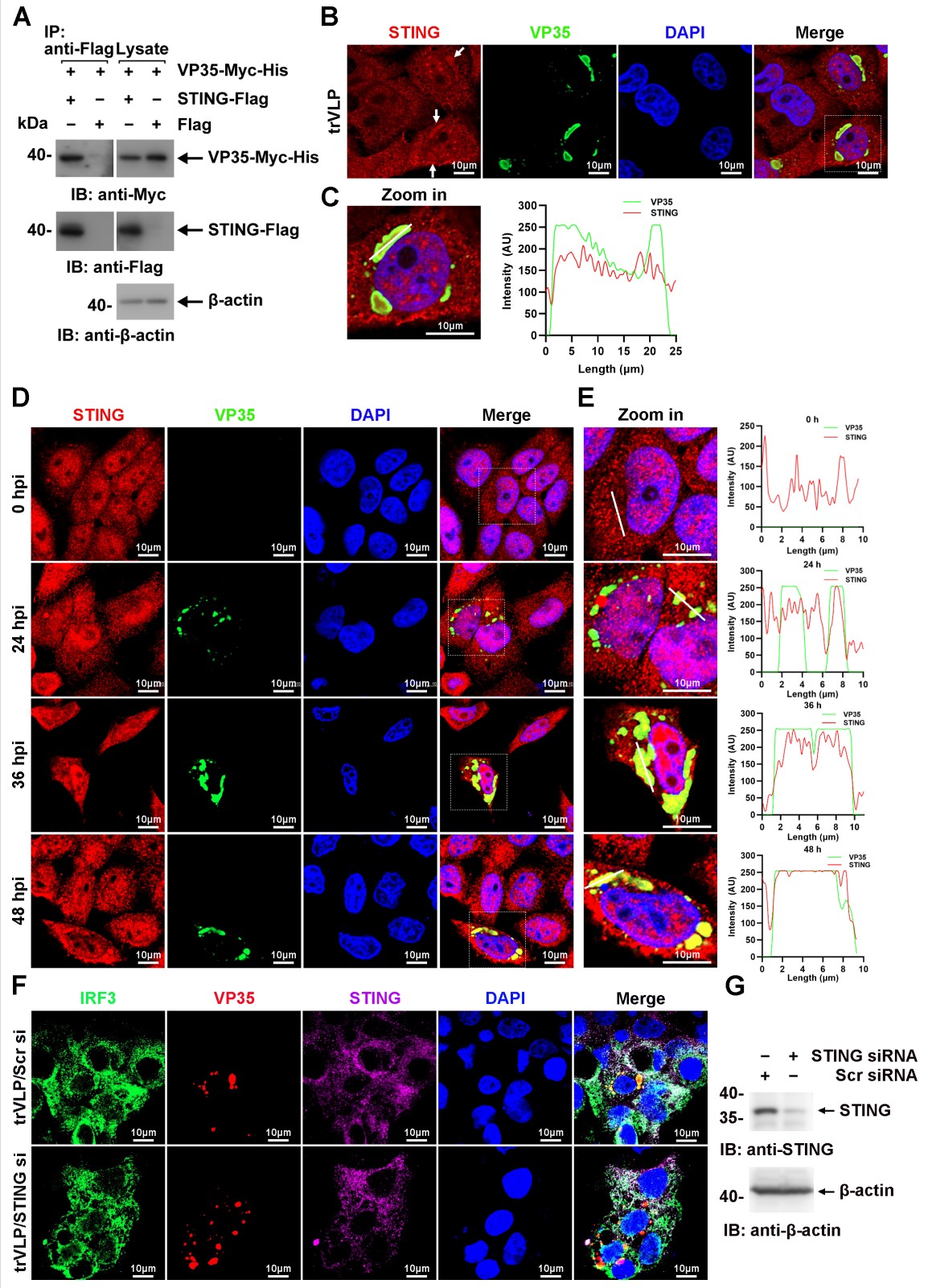

**Figure 6.** Ebola virus (EBOV) transcription- and replication-competent virus-like particles (trVLPs) recruit interferon regulatory factor 3 (IRF3) into viral inclusion bodies (IBs) via STING. (**A**) Lysates of HEK293 cells transfected with the indicated plasmids were subjected to anti-Flag immunoprecipitation and analyzed by immunoblotting. (**B**) HepG2 cells were transfected with the EBOV minigenome (p0). Forty-eight hours after infection, the cells were fixed and immunostained with anti-STING (red) and anti-VP35 (green) antibodies. White arrows: STING in IBs. Nuclei were stained with DAPI (blue), and

*Figure 6 continued on next page*

*Figure 6 continued*

images were obtained using a Zeiss LSM 800 Meta confocal microscope. Scale bar, 10 μm. (**C**) The left panel shows a magnified image of the IBs boxed in the merged panel of (**B**). The graphs (right panel) show the fluorescent intensity profiles along the indicated white lines drawn across one or more IBs. (**D**) HepG2 cells were infected with the EBOV trVLPs. At the indicated hours post infection (hpi), cells were fixed and immunostained with anti-STING (red) and anti-VP35 (green) antibodies. Nuclei were stained with DAPI (blue), and images were obtained using a Zeiss LSM 800 Meta confocal microscope. Scale bar, 10 μm. The data from two independent replicates are presented. (**E**) The left panel shows a magnified image of the IBs boxed in the merged panel of (**D**). The graphs (right panel) show fluorescent intensity profiles along the indicated white lines drawn across one or more IBs. (**F, G**) HepG2 cells were transfected with STING siRNA (STING si) or scrambled siRNA (Scr si) for 6 hr. The cells were then infected with the EBOV trVLPs for 36 hr and then immunostained with Fluor 488-conjugated-anti-IRF3 (green), anti-VP35 (red), and anti-STING (purple) antibodies. Nuclei were stained with DAPI (blue), and images were obtained using a Zeiss LSM 800 Meta confocal microscope. Scale bar, 10 μm. The silencing efficiency of STING siRNA was determined by immunoblotting (**G**).

The online version of this article includes the following source data and figure supplement(s) for figure 6:

**Source data 1.** Raw image for *Figure 6A*.

**Source data 2.** Numerical data for *Figure 6C*.

**Source data 3.** Numerical data for *Figure 6E*.

**Source data 4.** Raw image for *Figure 6G*.

**Figure supplement 1.** Ebola virus (EBOV) transcription- and replication-competent virus-like particles (trVLPs) recruit STING into viral inclusion bodies (IBs).

**Figure supplement 1—source data 1.** Numerical data for *Figure 6—figure supplement 1*.

of activated IRF3), but not IRF3, inhibited EBOV trVLPs replication in *IRF3*-depleted cells (*Figure 7G*). Importantly, compared with wild-type cells, *IRF3* depletion showed little, if any, effect on EBOV replication in the cells infected with live EBOV (*Figure 7H*). Taken together, these results suggest that the hijacking of IRF3 and sequestration into IBs by EBOV can be significantly more potent in the inhibition of IFN-I production and thereby antagonizes the inhibitory effect of IFN-I on viral replication.

## Discussion

Accumulating evidence suggests that EBOV has established multiple ways to antagonize host innate immune responses to maintain viral replication. Several EBOV proteins (VP35, VP24, GP, VP30, and VP40) are known to participate in host immune evasion to facilitate viral replication and pathogenesis (*Audet and Kobinger, 2015*; *Bhattacharyya, 2021*; *Cantoni and Rossman, 2018*). VP35 was demonstrated to suppress IFN-I production by inhibiting IRF3/7 phosphorylation, disrupting DC maturation, and facilitating the escape of immune sensation by dsRNA (*Basler, 2015*; *Cárdenas et al., 2006*; *Messaoudi et al., 2015*; *Prins et al., 2009*). VP30 and VP40 suppress RNA silencing by interacting with Dicer and modulating RNA interference components via exosomes, respectively (*Fabozzi et al., 2011*; *Pleet et al., 2017*). VP24 and GP are also known to block IFN-I signaling by hiding type I major histocompatibility complex (MHC-1) on the cell surface and counteracting tetherin or interfering with established immune responses by adsorbing antibodies against GP, respectively (*Audet and Kobinger, 2015*; *Bhattacharyya, 2021*).

Viral IBs are a characteristic of cellular EBOV infection and are important sites for viral RNA replication, and NP and VP35 are extremely critical proteins for the formation of IBs (*Hoenen et al., 2012*). However, whether viral IBs are involved in antagonizing IFN-I production during EBOV trVLPs infection has not yet been reported. Here, we found that IRF3 is hijacked and sequestered into EBOV IBs by viral infection (*Figure 1A*), which demonstrates that viral IBs are utilized for IRF3 compartmentalization. Meanwhile, this compartmentalization resulted in the spatial isolation of IRF3 from the kinases TBK1 and IKKε (*Figure 1C, E*). This suggests that IRF3 deprivation by viral IBs may antagonize host antiviral signaling by inhibiting IFN-I production signaling.

As expected, the expression of NP/VP35/VP30/L, which is involved in the composition of IBs, was significantly more antagonistic to SeV-induced IFN-β production than the expression of VP35 alone (*Figure 7B*). In addition, the expression of NP/VP35/VP30/L can significantly antagonize the promoting effect of poly(I:C) on IFN-β transcription, and IRF3 knockout could not further inhibit the transcription of IFN-β (*Figure 7C*), which may be because viral hijacking of IRF3 into IBs nearly completely antagonized its function of promoting IFN-β production. In this study, the effect of poly(I:C) is consistent with the results obtained with SeV, which indicates that poly(I:C) may mainly activate the RLR signaling

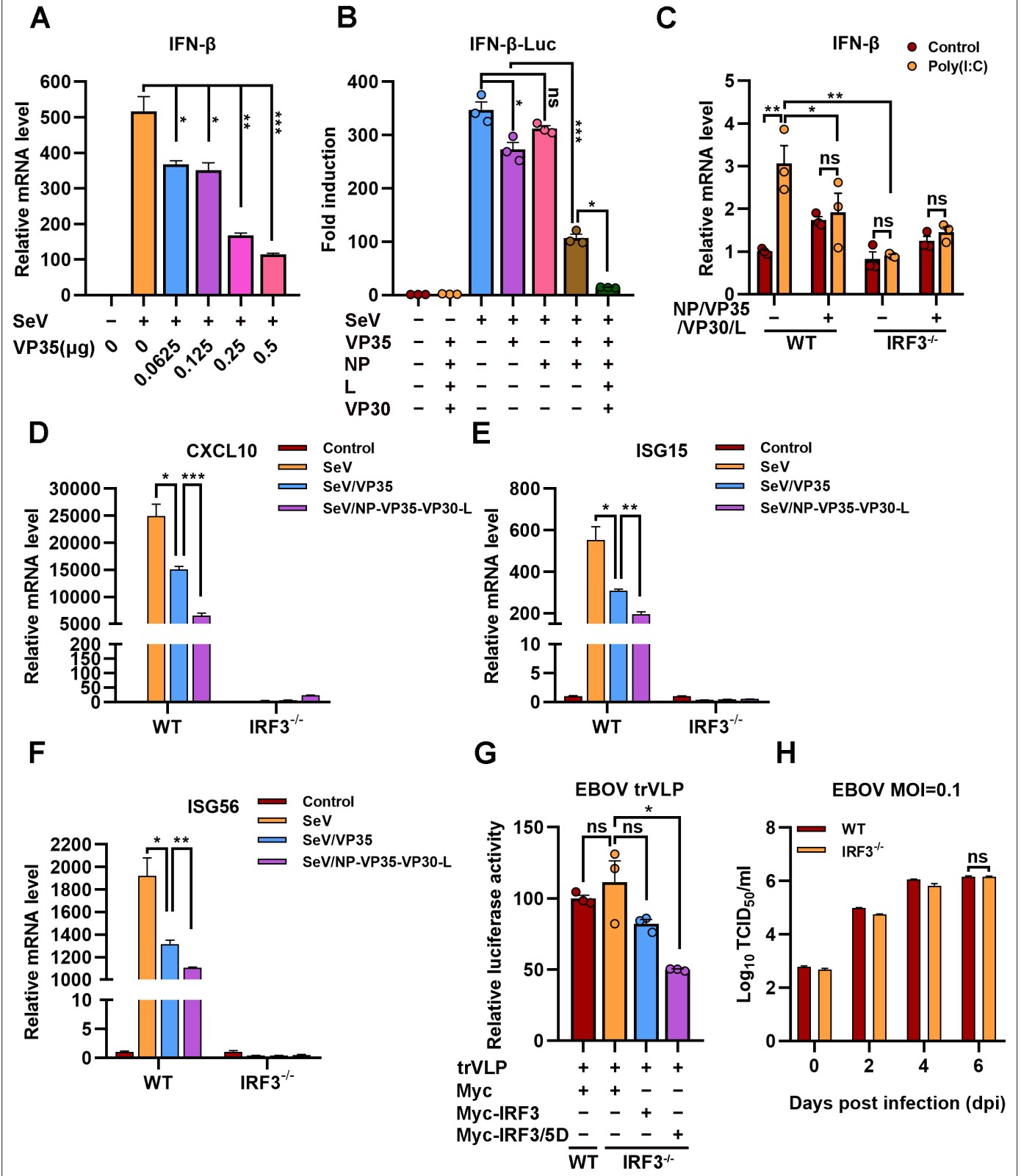

**Figure 7.** The hijacking of interferon regulatory factor 3 (IRF3) by viral inclusion bodies (IBs) inhibits IFN-β production. (**A**) HEK293 cells were transfected with the indicated plasmids for 24 hr, and the cells were infected with or without Sendai virus (SeV) at an MOI of 2 for another 12 hr. The mRNA level of IFN-β was quantified by quantitative RT-PCR (qRT-PCR). Differences between the two groups were evaluated by a two-sided unpaired Student's *t*-test. The data are presented as the means ±standard error of the mean (SEM; n=3; *p < 0.05, **p < 0.01, ***p < 0.001). (**B**) HEK293 cells were cotransfected

*Figure 7 continued on next page*

*Figure 7 continued*

with the firefly luciferase reporter plasmid pGL3-IFN-β-Luc, the *Renilla* luciferase control plasmid pRL-TK, and viral protein expression plasmids (0.0625 μg of pCAGGS-NP, 0.0625 μg of pCAGGS-VP35, 0.0375 μg of pCAGGS-VP30, and 0.5 μg of pCAGGS-L) for 24 hr, and the cells were infected with or without SeV at an MOI of 2 for another 12 hr. The luciferase activities were then analyzed. The data were analyzed to determine the fold induction by normalizing the firefly luciferase activity to the *Renilla* luciferase activity. Empty plasmid without SeV infection was used as a control, and the corresponding data point was set to 100%. Differences between the two groups were evaluated using a two-sided unpaired Student's *t*-test. The data are presented as the means ± SEM (n=3; ns, not significant, *$p < 0.05$, ***$p < 0.001$). (C) Wild-type (WT) and *IRF3*-depleted (IRF3$^{-/-}$) HeLa cells were transfected with or without pCASSG-NP, pCASSG-VP35, pCASSG-VP30, and pCASSG-L plasmids for 36 hr and then treated with or without 5 μg/ml poly(I:C) for 12 hr. The mRNA level of IFN-β was quantified by qRT-PCR. Differences between the two groups were evaluated using a two-sided unpaired Student's *t*-test. The data are presented as the means ± SEM (n=3; ns, not significant, *$p < 0.05$). (D–F) Wild-type (WT) and *IRF3*-depleted (IRF3$^{-/-}$) HeLa cells were transfected with or without pCAGGS-VP35 or pCASSG-NP, pCASSG-VP35, pCASSG-VP30, and pCASSG-L plasmids for 36 hr, and the cells were infected with or without SeV at an MOI of 5 for another 12 hr. The mRNA level of CXCL10 (D), ISG15 (E), and ISG56 (F) was quantified by qRT-PCR. Differences between the two groups were evaluated using a two-sided unpaired Student's *t*-test. The data are presented as the means ± SEM (n=3; *$p < 0.05$, **$p < 0.01$, ***$p < 0.001$). (G) Wild-type (WT) and *IRF3*-knockout (IRF3$^{-/-}$) HeLa cells were transfected with the Ebola virus (EBOV) minigenome (p0), pGL3-promoter and Myc-vector, Myc-IRF3 or Myc-IRF3/5D plasmids for 96 hr. The amounts of transcription- and replication-competent virus-like particles (trVLPs) were determined by a luciferase activity assay (left panel). Differences between the two groups were evaluated by a two-sided unpaired Student's *t*-test. The data are presented as the means ± SEM (n=3; ns, not significant, ***$p < 0.001$). (H) Wild-type (WT) and *IRF3*-knockout (IRF3$^{-/-}$) HeLa cells were infected with live EBOV (MOI = 0.1). The cell culture supernatants were collected on the indicated days post infection (dpi), and the viral titers were quantified as TCID$_{50}$ by a plaque assay. Differences between the two groups were evaluated using a two-sided unpaired Student's *t*-test. The data are presented as the means ± SEM (n=3; ns, not significant).

The online version of this article includes the following source data and figure supplement(s) for figure 7:

**Source data 1.** Numerical data for *Figure 7A*.

**Source data 2.** Numerical data for *Figure 7B*.

**Source data 3.** Numerical data for *Figure 7C*.

**Source data 4.** Numerical data for *Figure 7D*.

**Source data 5.** Numerical data for *Figure 7E*.

**Source data 6.** Numerical data for *Figure 7F*.

**Source data 7.** Numerical data for *Figure 7G*.

**Source data 8.** Numerical data for *Figure 7H*.

**Figure supplement 1.** The expression of interferon regulatory factor 3 (IRF3) and its mutants were detected by immunoblotting.

**Figure supplement 1—source data 1.** Raw image for *Figure 7—figure supplement 1A*.

**Figure supplement 1—source data 2.** Raw image for *Figure 7—figure supplement 1B*.

pathway (*Figure 3A*). As shown in *Figure 7D–F*, the expression of NP/VP35/VP30/L significantly inhibited the ability of SeV to promote the transcription of IFN-β downstream genes (CXCL10, ISG15, and ISG56) but did not completely suppress the effect of SeV, which may be due to the low transfection rate of HeLa cells. Furthermore, the knockout of IRF3 in cells could not further promote EBOV and EBOV trVLPs replication compared with that observed in wild-type cells (*Figure 7G, H*), which may have been because IRF3 was hijacked into viral IBs and could not be phosphorylated into the nucleus to regulate IFN-I production. These results suggest that viral IBs act as virus-built 'jails' to imprison transcription factors and present a novel and possible common mechanism of viral immune evasion in which the critical signaling molecule IRF3 is spatially segregated from the antiviral kinases TBK1 and IKKε.

Although almost all IRF3 could be sequestered to viral IBs formed by VP35 and NP (*Figure 5A, B*), we found that neither VP35 nor NP interacted with IRF3 (*Figure 5—figure supplement 1*). Here, we found that VP35 interacts with STING and colocalizes in IBs and that knockdown of STING inhibits the sequestration of IRF3 in IBs (*Figure 6A–G*). These results suggest that VP35 may hijack IRF3 into

*Figure 5 continued*

**Figure supplement 1—source data 2.** Raw image for *Figure 5—figure supplement 1B*.

**Figure supplement 2.** Neither VP24 nor VP30 plays an important role in sequestering interferon regulatory factor 3 (IRF3) into inclusion bodies (IBs).

**Figure supplement 2—source data 1.** Raw image for *Figure 5—figure supplement 2B*.

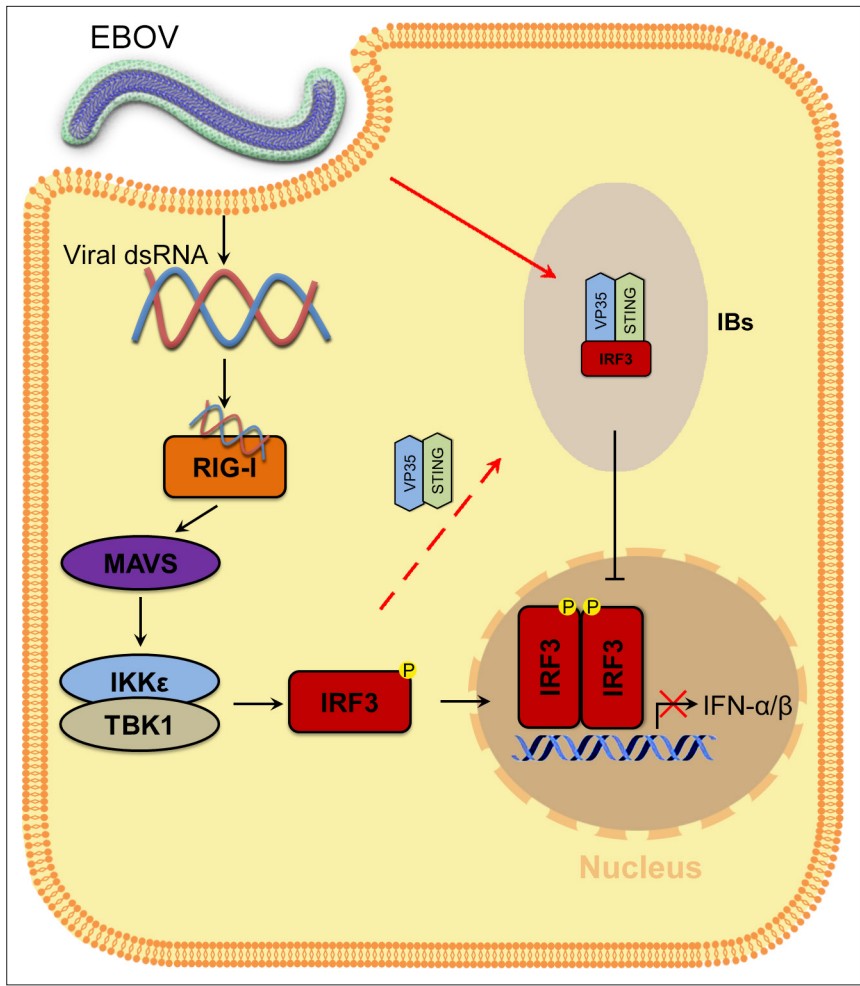

**Figure 8.** Model of the molecular mechanism by which EBOV hijacks IRF3 into viral IBs through VP35-STING to comprehensively disrupt IFN-I production. VP35 sequesters IRF3 to EBOV IBs, which in turn spatially segregates IRF3 from TANK-binding kinase 1 (TBK1) and I $\kappa$ B kinase epsilon (IKK$\epsilon$), blocks RLR signaling and inhibits IFN-I production.

IBs through STING. However, whether other host proteins are involved in this process and the role of NP in the recruitment of IRF3 by VP35 remain unclear. In addition, we found that VP35 may hijack IRF3 into IBs via STING association (*Figure 6A–C*); however, whether VP35 activates the STING-IRF3 pathway in a cGAS-independent manner by interacting with STING and the molecular mechanism remain to be further investigated.

In summary, EBOV VP35 sequesters IRF3 into viral IBs and inhibits the association of IRF3 with TBK1 and IKK$\epsilon$, preventing IRF3 from entering the nucleus and thereby inhibiting IFN-I production (*Figure 8*). Therefore, this study reveals a new strategy by which EBOV escapes the innate immune response and provides new ideas for EBOV disease treatment.

## Materials and methods

### Key resources table

| Reagent type (species) or resource | Designation | Source or reference | Identifiers | Additional information |
|---|---|---|---|---|
| Strain, strain background (*Escherichia coli*) | DH5α *E. coli* | TIANGEN | Cat# CB101 | Competent cells |

*Continued on next page*

*Continued*

| Reagent type (species) or resource | Designation | Source or reference | Identifiers | Additional information |
|---|---|---|---|---|
| Strain, strain background (*Sendai virus*) | SeV | ther | | Provided by Changchun Veterinary Research Institute |
| Strain, strain background (*Ebola virus*) | Ebola virus (Mayinga strain) | ther | | Provided by National Biosafety Laboratory, Chinese Academy of Sciences |
| Cell line (*Homo sapiens*) | IRF3-knockout HeLa cells (normal, adult) | Abclonal | Cat# RM02113 | |
| Antibody | anti-Flag M2 affinity Gel (Rabbit polyclonal) | Sigma-Aldrich | Cat# F2426; RRID:AB_2616449 | IP (1:50) |
| Antibody | anti-c-Myc affinity Gel (Mouse monoclonal) | Sigma-Aldrich | Cat# E6654; RRID:AB_10093201 | IP (1:50) |
| Antibody | HRP-conjugated anti-Flag antibody (Mouse monoclonal) | Sigma-Aldrich | Cat# A8592; RRID:AB_439702 | WB (1:4000) |
| Antibody | HRP-conjugated anti-c-Myc antibody (Mouse monoclonal) | Sigma-Aldrich | Cat# SAB4200742 | WB (1:2000) |
| Antibody | HRP-conjugated anti-β-Actin antibody (Mouse monoclonal) | Sigma-Aldrich | Cat# A3854; RRID:AB_262011 | WB (1:20,000) |
| Antibody | anti-Zaire Ebola virus VP35 antibody (Mouse monoclonal) | Creative Diagnostics | Cat# CABT-B292 | IF (1:50); WB (1:1000) |
| Antibody | anti-IRF3 antibody (Rabbit monoclonal) | Cell Signaling Technology | Cat# 11904; RRID:AB_2722521 | IF (1:50); WB (1:1000) |
| Antibody | anti-phospho-IRF3 (Ser396) antibody (Rabbit monoclonal) | Cell Signaling Technology | Cat# 29047; RRID:AB_2773013 | IB (1:500) |
| Antibody | anti-IRF3 antibody (Mouse monoclonal) | Cell Signaling Technology | Cat# 10949; RRID:AB_2797733 | IF (1:50) |
| Antibody | CoraLite Plus 488-conjugated IRF3 antibody (Rabbit polyclonal) | Proteintech | Cat# CL488-11312; RRID:AB_2919025 | IF (1:50) |
| Antibody | anti-TBK1 antibody (Rabbit monoclonal) | Abcam | Cat# ab40676; RRID:AB_776632 | IF (1:100) |
| Antibody | anti-TBK1 antibody (Rabbit monoclonal) | Cell Signaling Technology | Cat# 38066; RRID:AB_2827657 | IF (1:100) |
| Antibody | anti-IKKε antibody (Rabbit monoclonal) | Abcam | Cat# ab7891; RRID:AB_2124814 | IF (1:100) |
| Antibody | anti-STING antibody (Rabbit polyclonal) | Proteintech | Cat# 19851-1-AP; RRID:AB_10665370 | WB (1:1000) |
| Antibody | anti-STING antibody (Rabbit polyclonal) | Bioss | Cat# bs-8335R | IF (1:50) |
| Antibody | anti-Zaire Ebola virus NP antibody (Rabbit polyclonal) | Sino Biological | Cat# 40443-T62 | WB (1:1000) |
| Recombinant DNA reagent | Flag-VP35; Flag-NP (plasmid) | DOI: 10.1038/s41467-022-29948-4 | | |
| Recombinant DNA reagent | STING-Flag (plasmid) | Miaoling biology | Cat# P39762 | Flag-tagged of pCMV-vector (STING: NM_198282.4) |
| Recombinant DNA reagent | Flag-TBK1 (plasmid) | This paper | Synthesized by General Biol | Flag-tagged of pCDNA3.0-vector (TBK1: NM_013254.4) |

*Continued on next page*

*Continued*

| Reagent type (species) or resource | Designation | Source or reference | Identifiers | Additional information |
|---|---|---|---|---|
| Recombinant DNA reagent | Myc-IRF3 (plasmid) | This paper | Synthesized by General Biol | Myc-tagged of pCMV-vector (IRF3: NM_013254.4) |
| Recombinant DNA reagent | Myc-IRF3/5D (plasmid) | This paper | Synthesized by General Biol | The amino acids of IRF3 at S396, S398, S402, T404, and S405 were mutated to D |
| Recombinant DNA reagent | VP35-Myc-His (plasmid) | This paper | Synthesized by General Biol | Myc-tagged of pCMV-vector |
| Recombinant DNA reagent | pCAGGS-VP35; pCAGGS-NP; pCAGGS-VP30; pCAGGS-L; pCAGGS-T7; pCAGGS-Tim1; p4cis-vRNA-RLuc (plasmid) | DOI: 10.3791/52381 | | |
| Recombinant DNA reagent | pRL-TK vector (plasmid) | Promega | Cat# E2241 | |
| Recombinant DNA reagent | pGL3-IFNβ-Luc (plasmid) | This paper | | pGL3-basic vector |
| Recombinant DNA reagent | pGL3-Promoter (plasmid) | Youbio | Cat# VT1726 | |
| Sequence-based reagent | STING siRNA- sense | This paper | Synthesized by Tsingke Biotechnology | GCACCUGUGUCCUGGAGUATT |
| Sequence-based reagent | STING siRNA- anti-sense | This paper | Synthesized by Tsingke Biotechnology | UACUCCAGGACACAGGUGCTT |
| Sequence-based reagent | siRNA: nontargetin control-sense | This paper | Synthesized by Tsingke Biotechnology | UUCUCCGAACGUGUCACGUTT |
| Sequence-based reagent | siRNA: nontargetin control-anti-sense | This paper | Synthesized by Tsingke Biotechnology | ACGUGACACGUUCGGAGAATT |
| Sequence-based reagent | h-IFN-β-F | This paper | qPCR primers | AGGACAGGATGAACTTTGAC |
| Sequence-based reagent | h-IFN-β-R | This paper | qPCR primers | TGATAGACATTAGCCAGGAG |
| Sequence-based reagent | h-CXCL10-F | This paper | qPCR primers | TCCCATCACTTCCCTACATG |
| Sequence-based reagent | h- CXCL10-R | This paper | qPCR primers | TGAAGCAGGGTCAGAACATC |
| Sequence-based reagent | h-ISG15-F | This paper | qPCR primers | TCCTGGTGAGGAATAACAAGGG |
| Sequence-based reagent | h-ISG15-R | This paper | qPCR primers | CTCAGCCAGAACAGGTCGTC |
| Sequence-based reagent | h-ISG56-F | This paper | qPCR primers | TCGGAGAAAGGCATTAGATC |
| Sequence-based reagent | h-ISG56-R | This paper | qPCR primers | GACCTTGTCTCACAGAGTTC |
| Sequence-based reagent | h-GAPDH-F | This paper | qPCR primers | AAggTCATCCCTgAgCTgAAC |
| Sequence-based reagent | h-GAPDH-R | This paper | qPCR primers | ACgCCTgCTTCACCACCTTCT |
| Commercial assay or kit | ReverTra Ace qPCR RT Master Mix with gDNA Remover | TOYOBO | Cat# FSQ-301 | |
| Commercial assay or kit | SYBR Green Real-time PCR Master Mix | TOYOBO | Cat# QPK-201 | |

*Continued on next page*

*Continued*

| Reagent type (species) or resource | Designation | Source or reference | Identifiers | Additional information |
|---|---|---|---|---|
| Commercial assay or kit | Duolink in situ PLA reagent | Sigma-Aldrich | Cat# DUO92008 | |
| Commercial assay or kit | Dual-Luciferase Reporter Assay System | Promega | Cat# E1960 | |
| Software, algorithm | Prism 8.0 software | Graphpad | https://www.graphpad.com/scientific-software/prism/; | |
| Software, algorithm | ImageJ 1.48v software | National Institutes of Health | https://imagej.net/software/imagej/ | |
| Software, algorithm | QuantStudio 6 Flex multicolor real-time PCR Software | Applied Biosystems | | |
| Other | Mounting Medium with DAPI | Abcam | Ab104139 | DAPI is used for staining nuclei in immunofluorescence |

## Cell lines and transfections

HEK293, HeLa, and IRF3-knockout HeLa cells (ABclonal, RM02113) were grown in Dulbecco's modified Eagle's medium (Gibco). HepG2 cells were grown in minimum essential medium (Gibco) supplemented with a 1% nonessential amino acid solution (Gibco). All media were supplemented with 10% heat-inactivated fetal bovine serum (FBS; Gibco), 2 mM l-glutamine, 100 units/ml penicillin, and 100 units/ml streptomycin, and cells were grown at 37°C under an atmosphere with 5% $CO_2$. The cells were authenticated using short tandem repeat (STR) profiling and were also tested for mycoplasma contamination. Transient transfection was performed with Lipofectamine 3000 (Invitrogen) according to the manufacturer's instructions.

## Vectors and viruses

Flag-tagged VP35, NP, STING, and TBK1 vectors were constructed by cloning the corresponding gene fragments into a pcDNA3.0-based Flag-vector (Invitrogen). Myc-tagged VP35, IRF3, and IRF3/5D vectors were constructed by inserting the corresponding gene fragments into the pCMV-Myc vector (Clontech). All the constructs were validated by Sanger DNA sequencing.

SeV was amplified in 9- to 11-day embryonated specific pathogen-free eggs. Live EBOV (Mayinga strain) is preserved by the BSL-4 Lab at the Wuhan Institute of Virology, Chinese Academy of Sciences.

## Immunoprecipitation and immunoblot analysis

Cell lysates were prepared in lysis buffer containing 1% Nonidet P-40 and protease inhibitor cocktail (Roche) (*Cao et al., 2003*). Soluble proteins were immunoprecipitated using anti-Flag (M2, Sigma), anti-Myc (Sigma), or IgG of the same isotype from the same species as a negative control (Sigma). An aliquot of the total lysate (5%, vol/vol) was included as a control. Immunoblotting was performed with horseradish peroxidase (HRP)-conjugated anti-Myc (Sigma), HRP-conjugated anti-Flag (Sigma), HRP-conjugated anti-β-actin (Sigma), anti-VP35 (Creative Diagnostics), anti-IRF3 (Cell Signaling Technology), anti-phospho-IRF3 Ser396 (Cell Signaling Technology), anti-STING (Proteintech), or anti-NP (Sino Biological) antibodies. The antigen–antibody complexes were visualized via chemiluminescence (Immobilon Western Chemiluminescent HRP Substrate, Millipore). A PageRuler Western marker (Thermo) was used as a molecular weight standard.

## Gene silencing using siRNA

For gene knockdown in HepG2 cells, cells maintained in 6-well plates were transfected with 100 pmol STING siRNA (sense, 5′-GCACCUGUGUCCUGGAGUAUU-3′; antisense, 5′-UACUCCAGGACACAGGUGCUU-3′) or the same concentration of scrambled siRNA (sense, 5′-UCUCCGAACGUGUCACGUTT-3′; antisense, 5′-ACGUGACACGUUCGGAGAATT-3′) purchased from Tsingke Biotechnology (Beijing, China) with Lipofectamine 3000 (Invitrogen) according to the manufacturer's recommendations.

## Reverse transcription and quantitative RT-PCR

Total cellular RNA was prepared using an RNeasy Mini kit (QIAGEN, USA) according to the manufacturer's protocol. For cDNA synthesis, 0.5 µg of RNA was first digested with gDNA Eraser to remove contaminated DNA and then reverse transcribed using ReverTra Ace qPCR RT Master Mix with gDNA Remover (FSQ-301, Toyobo) in a 20-µl reaction volume. Then, 1 µl of cDNA was used as a template for quantitative PCR. The following primers were used in these experiments: h-IFN-β-F: 5′-AGGA CAGGATGAACTTTGAC-3′; h-IFN-β-R: 5′-TGATAGACATTAGCCAGGAG-3′; h-CXCL10-F: 5′-TCCC ATCACTTCCCTACATG-3′; h-CXCL10-R: 5′-TGAAGCAGGGTCAGAACATC-3′; h-ISG15-F: 5′-TCCT GGTGAGGAATAACAAGGG-3′; h-ISG15-R: 5′-CTCAGCCAGAACAGGTCGTC-3′; h-ISG56-F: 5′-TCGG AGAAAGGCATTAGATC-3′; h-ISG56-R: 5′-GACCTTGTCTCACAGAGTTC-3′; h-GAPDH-F: 5′-AAgg TCATCCCTgAgCTgAAC-3′; h-GAPDH-R: 5′-ACgCCTgCTTCACCACCTTCT-3′.

The samples were denatured at 95°C for 2 min, followed by 40 cycles of amplification (15 s at 94°C for denaturation, 60 s at 60°C for annealing and extension). Quantitative RT-PCR was performed using SYBR Green Real-time PCR Master Mix (QPK-201, Toyobo) with the QuantStudio 6 Flex multicolor real-time PCR detection system (ABI). Relative mRNA levels were normalized to GAPDH levels and calculated using the $2^{-\Delta\Delta CT}$ method (*Livak and Schmittgen, 2001*). The means (upper limit of the box) ± standard error of the mean (SEM; error bars) of three independent experiments are presented in the figures.

## In situ PLA

Duolink in situ PLA (Sigma) was used to detect the endogenous association of IRF3 and TBK1 in cells. In brief, HepG2 cells plated on glass coverslips were transfected with EBOV minigenome plasmids. After fixation with 4% formaldehyde, the cells were permeabilized with 0.3% Triton X-100 in phosphate-buffered saline (PBS) for 15 min. After blocking with blocking buffer (Sigma, DUO82007), the cells were incubated with mouse anti-IRF3 (Cell Signaling Technology) and rabbit anti-TBK1 (Abcam) primary antibodies. The nuclei were stained with DAPI (blue). The red fluorescent spots generated from the DNA amplification-based reporter system combined with oligonucleotide-labeled secondary antibodies were detected with a Zeiss LSM 800 Meta confocal microscope (Carl Zeiss).

## Immunofluorescence microscopy

Cells were transfected, fixed, permeabilized, and blocked as described above. Then, after incubation with anti-TBK1 (Cell Signaling Technology), anti-IKKε (Abcam), anti-IRF3 (Cell Signaling Technology), anti-VP35 (Creative Diagnostics), anti-NP (Sino Biological), or anti-STING (Bioss) antibodies overnight at 4°C, the cells were washed three times with phosphate buffered saline with tween 20 (PBST) buffer and then incubated with 488-conjugated anti-IRF3 (Proteintech) antibodies, FITC- or TRITC-conjugated goat anti-rabbit (or anti-mouse) IgG secondary antibodies for another 1 hr at room temperature. The cells were then stained with DAPI after washing and imaged using a laser scanning confocal microscope (Zeiss LSM 800 Meta) with a ×63 oil immersion lens.

## Luciferase reporter assay

The IFN-I production assay was performed as described previously (*Zhu et al., 2022*). Briefly, HEK293 cells ($1 \times 10^5$ cells per well in a 24-well plate) were cotransfected with the indicated amount of pCAGGS-NP (62.5 ng)/pCAGGS-VP35 (62.5 ng)/pCAGGS-VP30 (37.5 ng)/pCAGGS-L (500 ng), 200 ng of the IFN-β reporter plasmid (Promega, USA) and 4 ng of *Renilla* luciferase plasmid. An empty vector was used to ensure that each well contained the same plasmid concentration. After 24 hr, the cells were treated with SeV (MOI = 2) or 5 µg/ml poly(I:C) for 12 hr, and the luciferase activity of the cell lysates was analyzed with the dual-luciferase reporter assay system (Promega, E1960) using a GloMax 20/20 luminometer (Promega, USA). Values were obtained by normalizing the luciferase values to the Renilla values. Fold induction was determined by setting the results from the group transfected with vector without Flag-VP35 to a value of 1.

## EBOV trVLPs assay

The replication of EBOV in the cells was evaluated with the minigenome system (*Hoenen et al., 2014*). Briefly, producer cells (p0) were cotransfected with p4cis-vRNA-RLuc (250 ng) and pCAGGS-T7 (250 ng) for T7 RNA polymerase expression and four plasmids for EBOV protein expression (pCAGGS-NP

(125 ng), pCAGGS-VP35 (125 ng), pCAGGS-VP30 (75 ng), and pCAGGS-L (1000 ng)), as well as the luciferase reporter vector pGL3-Promoter (Youbio, 25 ng). One day after transfection, the medium was replaced with medium containing 5% FBS, and the cells were then incubated for another 3 days. Viral replication was determined by intracellular luciferase activities using a dual-luciferase reporter assay kit (Promega, E1960) after cell lysis with passive lysis buffer (PLB, Promega). For immunofluorescence experiments, cells were harvested 48 hr after transfection.

### Transmission electron microscopy

HepG2 cells transfected with EBOV minigenome p0-related plasmids were washed with PBS, fixed with 2.5% glutaraldehyde, and then prestained with osmium tetroxide. Eighty-nanometer-thick serial sections were then cut and stained with uranyl acetate and lead citrate. Images were acquired with a transmission electron microscope (Hitachi, H-7650) operating at 80 kV.

### EBOV infection assay

HepG2, HeLa, or *IRF3*-depleted HeLa cells grown to ~70% confluency in 12-well plates (for viral proliferation) or 12-well plates with a 18-mm coverslip (for immunofluorescence microscopy) were incubated with the EBOV Mayinga strain, which was tittered in Vero E6 cells, at 37°C for 1 hr at the indicated MOI. Then, the cells were washed three times with PBS, and fresh medium was added to the cells, which were incubated at 37°C for 72 hr (for microscopy) or the indicated times (0, 2, 4, and 6 days; for the viral proliferation assay). Subsequently, the cells on the coverslip were fixed with 4% formaldehyde for immunofluorescent straining, and the supernatants were collected at the indicated times for viral titration following the requirements of the BSL-4 laboratory. The viral titers were determined by plaque formation assay. Briefly, 10-fold serially diluted samples (100 µl) were added to 96-well plates containing $1 \times 10^4$ Vero E6 cells per well and incubated for 1 hr at 37°C in a 5% $CO_2$ incubator. Then, 100 µl of medium containing 2% FBS was added to each well. After incubation for 5–7 days at 37°C in a 5% $CO_2$ incubator, the cytopathic effect was observed, and the median tissue culture infective dose ($TCID_{50}$)/ml was calculated. All work with live EBOV was performed with BSL-4 containment.

### Statistical analyses

Graphical representation and statistical analyses were performed using Prism 8 software (GraphPad Software). Unless indicated otherwise, the results are presented as the means (upper limit of the box) ± SEM (error bars) from three independent experiments conducted in duplicate. An unpaired two-tailed *t*-test was used for the analysis of two groups. Data were considered significant when $p < 0.05$ (*), $p < 0.01$ (**), and $p < 0.001$ (***).

## Acknowledgements

This work was supported by the National Natural Science Foundation of China (82372255), the Advanced Customer Cultivation Project of the Wuhan National Biosafety Laboratory, the Chinese Academy of Sciences [2022ACCP-MS04], and the National Major Science and Technology Projects of China [2018ZX09711003-005-005 to TG and 2022YFC2600704 to HL].

## Additional information

### Funding

| Funder | Grant reference number | Author |
| --- | --- | --- |
| National Natural Science Foundation of China | 82372255 | Lin Zhu |
| Advanced Customer Cultivation Project of the Wuhan National Biosafety Laboratory, the Chinese Academy of Sciences | 2022ACCP-MS04 | Cheng Cao |

| Funder | Grant reference number | Author |
| --- | --- | --- |
| National Major Science and Technology Projects of China | 2018ZX09711003-005-005 | Ting Gao |
| National Major Science and Technology Projects of China | 2022YFC2600704 | Hainan Liu |

The funders had no role in study design, data collection, and interpretation, or the decision to submit the work for publication.

## Author contributions

Lin Zhu, Conceptualization, Resources, Data curation, Software, Formal analysis, Supervision, Funding acquisition, Validation, Investigation, Writing – original draft, Writing – review and editing; Jing Jin, Resources, Data curation, Software, Formal analysis, Validation; Tingting Wang, Resources, Data curation, Formal analysis, Investigation, Methodology; Yong Hu, Software, Formal analysis, Investigation, Methodology; Hainan Liu, Software, Formal analysis, Funding acquisition, Investigation; Ting Gao, Formal analysis, Supervision, Funding acquisition, Investigation, Visualization; Qincai Dong, Resources, Formal analysis; Yanwen Jin, Formal analysis, Supervision, Investigation; Ping Li, Formal analysis, Supervision; Zijing Liu, Resources, Supervision; Yi Huang, Resources, Data curation, Project administration; Xuan Liu, Supervision, Investigation, Project administration, Writing – review and editing; Cheng Cao, Conceptualization, Supervision, Funding acquisition, Validation, Investigation, Visualization, Project administration, Writing – review and editing

## Author ORCIDs

Lin Zhu ![ORCID] https://orcid.org/0000-0001-5772-9482
Yi Huang ![ORCID] https://orcid.org/0000-0001-6585-5243

Reviewer #3 (Public Review): https://doi.org/10.7554/eLife.88122.3.sa1
Reviewer #4 (Public Review): https://doi.org/10.7554/eLife.88122.3.sa2
Author Response https://doi.org/10.7554/eLife.88122.3.sa3

---

# Additional files

## Supplementary files

- MDAR checklist

## Data availability

All data generated or analyzed during this study are included in the manuscript and supporting files. Source data are provided within this article.

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
