## [Editor Report · eLife assessment]

This study explores how Ebola virus evades human immune responses. The study reports a potential new mechanism wherein Ebola virus traps human IRF3, a key transcription factor involved in immune signaling, into virus-produced "inclusion bodies". The topic is **important**, the paper has many merits, and the biochemical assays are **solid**. However, the current data do not clearly explain the relationship between the VP35 protein and IRF3.

---

## [Referee Report · Reviewer #3 (Public Review)]

Summary:

In the manuscript "Ebola Virus Sequesters IRF3 in Viral Inclusion bodies to Evade Host Antiviral Immunity " by Lin Zhu et al, the authors elucidated an evasion mechanism by which EBOV evades host innate immunity.

Strengths:

Using data from immunofluorescence analysis, TEM and Western Blot, the authors conclude that Ebola virus VP35 protein evades host antiviral immunity by interacting with STING to sequester IRF3 into IBs and inhibit type-I interferon production.

Weaknesses:

Similar mechanisms have already been found in other viruses, such as SFTSV, RSV and so on. In addition, the presented results are also relatively rough, and the mechanism explained is not deep enough, so this story is not innovative

---

## [Referee Report · Reviewer #4 (Public Review)]

The manuscript entitled "Ebola Virus Sequesters IRF3 in Viral Inclusion Bodies to Evade Host Antiviral Immunity" mainly describes that the function of IBs formed by the viral proteins VP35 and NP in evading host antiviral immunity. They proved that Ebola virus VP35 protein can interact with STING, but not IRF3, to sequester IRF3 into inclusion bodies and thereby inhibit type-I interferon production. This work will be of some interest to readers in the Ebola Virus field, however, the current data do not clearly explain the relationship of VP35 protein and IRF3.

---

## [Author Response]

The following is the authors’ response to the original reviews.

**Public Reviews:**

**Reviewer #1 (Public Review):**
The authors were trying to investigate whether viral IBs are involved in antagonizing IFN-I production during EBOV trVLPs infection. They found that IRF3 is hijacked and sequestered into EBOV IBs after viral infection, thereby leading to the spatial isolation of IRF3 with TBK1 and IKKε. In such a progress, the activity of IRF3 is suppressed and downstream IFN-I induction is inhibited. The authors designed many experiments, such as the PLA that examined the colocalization, to support their conclusions. However, necessary negative controls were missed in several assays. More key index is needed to be examined in several assays.The paper is well organized and most data in this paper could support the conclusions, while there are several issues that need to be further solved.1. In Figure 2-4, authors should examine the expression of downstream IFNs as well as the phosphorylation and nuclear localization of IRF3 to further prove the suppression of IRF3 activity by infecting with trVLPs.

Response: The inhibitory effect of trVLPs infection on the phosphorylation of IRF3 S396 and SeV-induced IRF3 nuclear localization was determined by immunoprecipitation (Figure 3D) and immunofluorescence (Figure 4A and 4B), respectively. In addition, we demonstrated that IFN-β transcription was inhibited more potently by EBOV viral inclusion bodies compared with VP35 alone (Figure 7B and 7C).

Moreover, EBOV viral inclusion bodies were demonstrated to inhibit the transcription of IFN downstream genes (e.g., CXCL10, ISG15 and ISG56) more potently than VP35 alone (new Figure 7D-F).

1. In Figure 5, to better prove the conclusion that EBOV NP and VP35 play an important role in sequestering IRF3 in IBS, authors should add the "NP+VP35+VP30" and "NP+VP35+VP24" groups to reperform the assay.

Response: According to the reviewer’s suggestion, VP24 or VP30 was added to the “VP35+NP” group, and the results showed that the “NP+VP35+VP24” and “NP+VP35+VP30” groups exhibited little, if any, effect on the distribution of IRF3 compared with the “NP+VP35” group (new Figure 5 - figure supplement 2A-B).

1. In Figure 6f, the expression of STING should be examined by immunostaining to show the knockdown efficiency in trVLPs-infected cells.

Response: As suggested by the reviewer, immunostaining was performed to visually detect the effect of STING knockdown on the IRF3 distribution during trVLPs infection (new Figure 6F).

**Reviewer #2 (Public Review):**
The manuscript by Zhu et al explored molecular mechanisms by which Ebola virus (EBOV) evades host innate immune response. EBOV has a number of means to shut down the type I interferon induction (by viral VP35 protein) and block type I interferon action (by viral VP24 protein). This study reported a new mechanism that inclusion body (IB) used for viral replication sequesters IRF3, a key transcription factor involved in the interferon signaling, resulting in blockade of downstream type I interferon gene transcription. This finding is potentially interesting and may provide a new insight into EBOV's evasion of innate immunity. However, there are some flaws in the experimentations and analyses that need to be addressed.1. Most of experiments were performed by transfection of trVLP plasmids, which is very different from virus infection. The conclusions should be examined and verified in the context of virus infection.

Response: As suggested by the reviewer, the effects of IRF3 depletion on live Ebola virus replication were examined as described in the revised manuscript. Consistent with the results obtained after trVLPs infection, IRF3 depletion exerted little, if any, effect on viral replication (new Figure 7H), which supports the notion that, upon EBOV infection and the formation of inclusion bodies, IRF3 has little, if any, transcription activation activity after sequestration by inclusion bodies.

1. Fig 1 - VP35 displayed a classical IB staining only in Panel A, while much less so in Panel C and not in panel B. It seemed that the VP35 staining images were chosen in a way towards the authors' favor. The statistical analysis of co-localization of VP35 and IRF3, TBK1 or IKKe should be performed to draw the conclusion. Another concern is that IKKe is normally lowly expressed under a rest condition and becomes induced only when the interferon signaling is activated. It seemed to be expressed at a high level even when the interferon signaling is blocked in Panel C. The authors should comment on this discrepancy.

Response: Ebola virus inclusion bodies show variations in both shape and size. According to the reviewer’s suggestion, the colocalization of TBK1 or IKKε and VP35 is shown in new figures (new Figure 1C and 1E), and quantitatively analyzed by the fluorescence intensity using ImageJ software (new Figure 1B, 1D and 1F).

1. Fig 2 - Was this experiment done by transfection or infection? The description of result is not consistent with the figure legend. The labeling was also not consistent between panel A and B. I would suggest performing Western blot to analyze the expression level of IRF3.

Response: We apologize for the incorrect description of the data. Ebola virus trVLPs were initially produced based on transfection but also involved the viral infection process. The use of “transfection” in the figure and figure legends has been changed to “infection” in the revised manuscript.As suggested by the reviewer, Western blotting was performed to analyze the IRF3 expression levels at different time points after trVLPs infection (new Figure 2D).

1. Fig 3 and 4 - As VP35 is well known for its highly efficient blockade of type I interferon activation, how would the authors differentiate the effect of VP35 alone from the sequestration of IRF3 in IBs in these experiments?

Response: Previous studies have found that VP35, rather than NP, inhibits the expression of interferon, and the “VP35+NP” treatment, which induces IRF3 sequestration, showed inhibited IFN-β luciferase activity much more potently than VP35 expression alone (Figure 7B).

1. Fig 3 - PolyIC can activate both RLR and TLR signaling pathways. Can the author comment on which pathway it activates in this experiment?

Response: In this study, the effect of poly(I:C) was consistent with the results observed with SeV, which indicated that poly(I:C) may mainly activate the RLR signaling pathway. A discussion was added to the revised manuscript.

1. The authors demonstrated that VP35 interacts with STING and recruit the latter to IBs. How would this affect the function of STING given that STING plays essential roles in cGAS/cGAMP pathway?

Response: This study unexpectedly showed that VP35 can recruit IRF3 into viral inclusion bodies through STING, but whether it regulates the cGAS-STING pathway remains to be further investigated. Related discussion was added to the revised manuscript.

1. It is difficult to follow the logics of Fig 7. The expression level of each viral protein should be determined. Ideally, a mutation in VP35 that disrupts its ability to antagonize the interferon signaling but still allows for the IB formation can be used to assess the relative contribution of IB sequestering IRF3.

Response: As suggested by the reviewer, a series of VP35 mutants were constructed, but we failed to obtain a VP35 mutant that contains a mutation that disrupts the ability of the protein to antagonize interferon signaling but still allows IB formation. Instead, coexpression of “NP+VP35+VP30+L”, which induces IBs formation, inhibited IFN-I more potently than the expression of VP35 alone (Figure 7B). IRF3 knockout inhibited poly(I:C)-induced IFN-I production but had little, if any, effect on poly(I:C)-induced IFN-I production in the “NP+VP35+VP30+L” group (Figure 7C). IRF3 knockout in the cells did not significantly affect viral replication, but overexpression of activated IRF3 (IRF3/5D), instead of wild-type IRF3, inhibited viral replication (new Figure 7G-H). These results collectively suggested that almost all IRF3 in cells was hijacked and sequestered into IBs in the Ebola virus-infected cells.